# Oridonin Attenuates *Burkholderia cenocepacia* Virulence by Suppressing Quorum-Sensing Signaling

Xia Li,[a] Kai Wang,[b] Gerun Wang,[a] Binbin Cui,[a] Shihao Song,[a] Xiuyun Sun,[a] ⬦Yinyue Deng[a]

[a]School of Pharmaceutical Sciences (Shenzhen), Shenzhen Campus of Sun Yat-sen University, Sun Yat-sen University, Shenzhen, China
[b]College of Agriculture, South China Agricultural University, Guangzhou, China

Xia Li, Kai Wang, and Gerun Wang contributed equally to this article. Author order was determined by their equal but gradated contributions for this paper.

**ABSTRACT** *Burkholderia cenocepacia* is a human opportunistic pathogen that mostly employs two types of quorum-sensing (QS) systems to regulate its various biological functions and pathogenicity: the *cis*-2-dodecenoic acid (BDSF) system and the *N*-acyl homoserine lactone (AHL) system. In this study, we reported that oridonin, which was screened from a collection of natural products, disrupted important *B. cenocepacia* phenotypes, including motility, biofilm formation, protease production, and virulence. Genetic and biochemical analyses showed that oridonin inhibited the production of BDSF and AHL signals by decreasing the expression of their synthase-encoding genes. Furthermore, we revealed that oridonin directly binds to the regulator RqpR of the two-component system RqpSR that dominates the above-mentioned QS systems to inhibit the expression of the BDSF and AHL signal synthase-encoding genes. Oridonin also binds to the transcriptional regulator CepR of the *cep* AHL system to inhibit its binding to the promoter of *bclACB*. These findings suggest that oridonin could potentially be developed as a new QS inhibitor against pathogenic *B. cenocepacia*.

**IMPORTANCE** *Burkholderia cenocepacia* is an important human opportunistic pathogen that can cause life-threatening infections in susceptible individuals. It employs quorum-sensing (QS) systems to regulate biological functions and virulence. In this study, we have identified a lead compound, oridonin, that is capable of interfering with *B. cenocepacia* QS signaling and physiology. We demonstrate that oridonin suppressed *cis*-2-dodecenoic acid (BDSF) and *N*-acyl homoserine lactone (AHL) signal production and attenuated virulence in *B. cenocepacia*. Oridonin also impaired QS-regulated phenotypes in various *Burkholderia* species. These results suggest that oridonin could interfere with QS signaling in many *Burkholderia* species and might be developed as a new antibacterial agent.

**KEYWORDS** *Burkholderia cenocepacia*, quorum sensing, BDSF, AHL, oridonin

**B**urkholderia cenocepacia is a Gram-negative opportunistic pathogen that causes life-threatening infections in immune-deficient individuals, particularly in patients with chronic granulomatous diseases and cystic fibrosis (CF) (1, 2). *B. cenocepacia* usually employs two types of quorum-sensing (QS) systems: the *cis*-2-dodecenoic acid (BDSF) system and the *N*-acyl homoserine lactone (AHL) system (3). BDSF is synthesized by the bifunctional crotonase RpfF$_{BC}$ (Bcam0581), which is a highly conserved protein in *Burkholderia cepacia* complex members (4–6). In the BDSF system, BDSF activates the cyclic diguanosine monophosphate (c-di-GMP) phosphodiesterase activity of RpfR through binding to its PAS domain, which leads to a decrease in the intracellular c-di-GMP level to then regulate the transcription of target genes through the RpfR-GtrR complex (3). The AHL synthase CepI mainly synthesizes *N*-octanoyl-homoserine lactone (C8-HSL) as the major AHL signal and a small amount of *N*-hexanoyl-homoserine lactone (C6-HSL) (7). When AHL accumulates in the environment and reaches a signal

Address correspondence to Yinyue Deng, dengyle@mail.sysu.edu.cn.

The authors declare no conflict of interest.

concentration threshold, it can bind to the CepR protein to form a complex, which activates or inhibits the expression of target genes by binding to promoters (7). The above-mentioned two systems have overlapping effects on various biological functions, including motility, biofilm formation, and virulence factor production (8). Recently, a two-component system, RqpSR, was found to control the production of BDSF and AHL signals by regulating the transcriptional expression levels of signal synthase-encoding genes in *B. cenocepacia* (9).

Antibiotics have been extensively used to prevent and control infectious diseases caused by pathogenic microorganisms. However, antibiotic abuse has fostered the emergence of superbugs and caused a severe public health threat (10). The development of antibiotic resistance in *B. cenocepacia* has become a serious issue. *B. cenocepacia* can induce chromosomal $\beta$-lactamases and alter penicillin-binding proteins; in addition, this pathogen also possesses an antibiotic efflux pump, resulting in resistance to chloramphenicol, quinolones, and trimethoprim (11), which increases the difficulty of controlling clinical infections (12). Recently, inhibition of bacterial virulence has become a potentially effective method to combat bacterial infection rather than to simply control bacterial growth. This approach reduces the pressure on the bacteria to survive and effectively slows the spread of drug resistance (13, 14). Because QS is widely utilized by many pathogens to regulate virulence (15), the design and development of new antimicrobial strategies targeting QS might be an attractive method to control the virulence of bacterial pathogens.

Oridonin is a bioactive ent-kaurane diterpenoid isolated from *Isodon rubescens* that is commonly used in traditional Chinese medicine (16). In recent years, the potential role of oridonin in cancer treatment has attracted increasing attention (17). In addition, oridonin exhibits anti-inflammatory activity. Oridonin suppresses the release of proinflammatory cytokines, such as tumor necrosis factor-$\alpha$ (TNF-$\alpha$) and interleukin-6 (IL-6), by inhibiting NF-$\kappa$B or mitogen-activated protein kinase (MAPK) activation (18). Both the AHL and BDSF systems are conserved QS systems in *Burkholderia* species that regulate various biological functions and virulence (7, 19). Therefore, the AHL and BDSF systems are attractive potential candidate targets to be used to inhibit the virulence of bacteria. In our previous study, we reported that a QS signal inhibitor (*cis*-14-methylpentadec-2-enoic acid) showed strong interference with BDSF signaling and virulence but did not inhibit the growth rate of *B. cenocepacia* cells (20, 21). In this study, we screened and evaluated 1,000 natural products for their ability to inhibit *B. cenocepacia* biofilm formation and virulence. One lead compound, oridonin, showed excellent efficacy in interfering with *B. cenocepacia* QS signaling and attenuating virulence but did not noticeably influence the growth rate, suggesting that it could potentially be developed as a novel antimicrobial agent against *B. cenocepacia* infection.

## RESULTS

**Screening of the leading compounds to inhibit the virulence of *B. cenocepacia*.** The lectin-encoding *bclACB* operon is related to biofilm formation, which plays an important role in the pathogenicity of *B. cenocepacia* (8). To screen for effective antivirulence compounds, we first tested the effects of approximately 1,000 natural compounds on *B. cenocepacia* H111 carrying a *bclACB-gfp-lacZ* promoter fusion plasmid by fluorescence microscopy at a final concentration of 20 $\mu$M (data not shown). Then, 113 active candidate compounds were selected and tested for their efficacy to inhibit *B. cenocepacia* biofilm formation (Fig. S1). Next, we tested whether these compounds influence *B. cenocepacia* virulence using an A549 cell line infection model. Cytotoxicity was determined by measuring the amount of lactate dehydrogenase (LDH) released into the supernatant of cultured A549 cells. The results showed that only 13 compounds effectively reduced the virulence of *B. cenocepacia* by more than 10% (Fig. 1), among which theaflavin-3,3′-digallate, (+)-brazilin, theaflavin-3-gallate, $\beta$-hydroxylsovalerylshikonin, oridonin, and 3′-hydroxypterostilbene exhibited notable inhibitory activities to reduce the virulence of *B. cenocepacia* by more than 30% (Fig. 1).

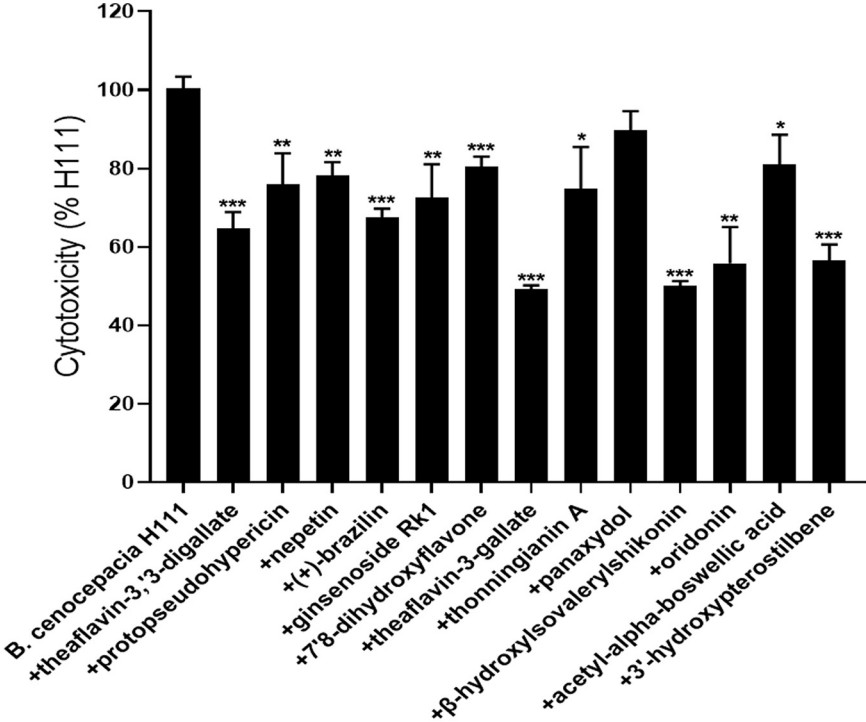

**FIG 1** Effects of the compounds on the virulence of *B. cenocepacia* H111. Cytotoxicity was determined by measuring lactate dehydrogenase (LDH) release. Compounds were dissolved in dimethyl sulfoxide (DMSO), and the same volume of DMSO that was used as the solvent for the compounds was used as a control. The amount of LDH released by A549 cells after inoculation with *B. cenocepacia* H111 in the presence of DMSO without compounds was defined as 100% to normalize the LDH release ratios of the other treatments. The data are presented as the means ± standard deviations of three independent experiments. The significance of the results was determined by one-way analysis of variance (ANOVA). *, $P < 0.05$; **, $P < 0.01$; ***, $P < 0.001$.

**Identification of the potential QS inhibitors in *B. cenocepacia*.** The *bclACB* operon is controlled by both the BDSF and AHL systems (3), so we tested the effects of the above 13 compounds on the BDSF and AHL systems, as they were selected for their effective inhibition of the P*bclACB*-*gfp*-*lacZ* reporter, biofilm formation and cytotoxicity. The P*rpfF*$_{BC}$-*lacZ* and P*cepI*-*lacZ* reporter strains were used to measure the effects of these compounds on the BDSF and AHL systems. The results showed that theaflavin-3,3′-digallate, thonningianin A, acetyl-$\alpha$-boswellic acid, 3′-hydroxypterostilbene, and oridonin significantly reduced *rpfF*$_{BC}$ gene expression at a final concentration of 20 $\mu$M (Fig. 2A). Additionally, theaflavin-3,3′-digallate, (+)-brazilin, ginsenoside Rk1, thonningianin A, $\beta$-hydroxylsovalerylshikonin, oridonin, and acetyl-$\alpha$-boswellic acid significantly inhibited *cepI* gene expression at a final concentration of 20 $\mu$M (Fig. 2B).

**Oridonin inhibits QS in *B. cenocepacia* by directly binding to RqpR.** The two-component system RqpSR controls BDSF and AHL signal production by directly regulating the transcriptional levels of signal synthase-encoding genes (9). As theaflavin-3,3′-digallate, thonningianin A, oridonin, and acetyl-$\alpha$-boswellic acid exhibited obvious inhibition on the expression of *rpfF*$_{BC}$ and *cepI* (Fig. 2), to further study whether these compounds affect the expression of *rpfF*$_{BC}$ and *cepI* through RqpR, we purified the RqpR protein to perform isothermal titration calorimetry (ITC) analysis (Fig. S2; Fig. 3A). It was found that only RqpR bound to oridonin with an estimated dissociation constant ($K_D$) of 8.28 ± 0.895 $\mu$M. To further explore the relationship between oridonin and RqpR regulatory activity, we then performed electrophoretic mobility shift assays (EMSAs) to determine whether oridonin affects RqpR binding to the promoter DNA of target genes. As shown in Fig. 3B and C, the binding of RqpR to the *rpfF*$_{BC}$ and *cepI* promoter probes was inhibited when oridonin was present in the reaction mixtures, and the amount of probe bound to RqpR decreased with increasing oridonin concentrations.

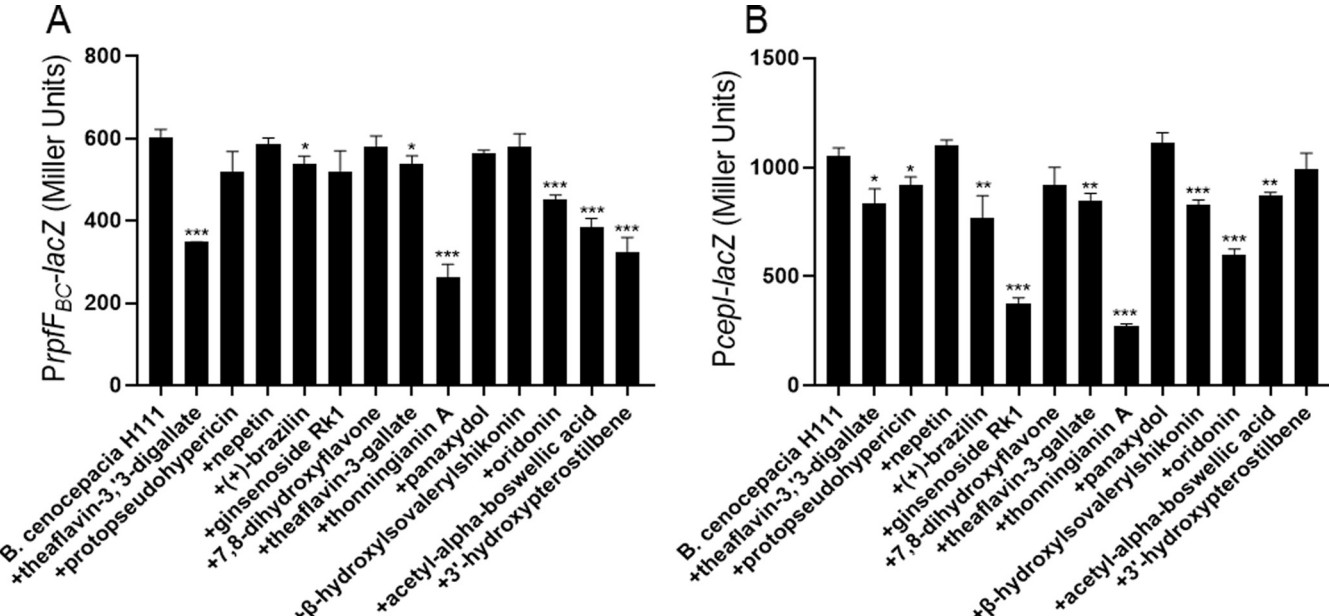

**FIG 2** Influences of the candidate compounds on the quorum-sensing (QS) systems in *B. cenocepacia* H111. (A) Effects of exogenous addition of 20 $\mu$M candidate compounds on *rpfF_{BC}* gene expression as determined by assessing the $\beta$-galactosidase activity of the P*rpfF_{BC}-lacZ* transcriptional fusions. (B) Effects of exogenous addition of 20 $\mu$M candidate compounds on *cepI* gene expression as determined by assessing the $\beta$-galactosidase activity of the P*cepI-lacZ* transcriptional fusions. Compounds were dissolved in DMSO, and the same volume of DMSO that was used as the solvent for the compounds was used as a control. The data are presented as the means $\pm$ standard deviations of three independent experiments. The significance of the results was determined by one-way ANOVA. *, $P < 0.05$; **, $P < 0.01$; ***, $P < 0.001$.

We next explored whether oridonin inhibits BDSF signal synthesis. As shown in Fig. 3D, the exogenous addition of 20, 50, or 100 $\mu$M oridonin to the wild-type strain resulted in a decrease in *rpfF_{BC}* expression by 21.39, 48.68, and 64.43%, respectively. Moreover, oridonin reduced BDSF production in a dose-dependent manner (Fig. 3E). Interestingly, exogenous addition of oridonin at final concentrations of 20 to 100 $\mu$M did not affect the growth of *B. cenocepacia* (Fig. S3).

Oridonin exerts an inhibitory effect on the transcriptional expression levels of both BDSF and AHL signal synthesis-encoding genes (Fig. 2). The BDSF system positively regulates *cepI* expression and AHL signal production (22); therefore, we next investigated the effect of oridonin on AHL signal synthesis. As shown in Fig. 3F, *cepI* activity in the wild-type strain decreased by 71% when treated with oridonin at a final concentration of 100 $\mu$M. Consistent with the downregulated activity of the *cepI* promoter-*lacZ* fusion, AHL signal production was reduced by 19.5, 33.3, and 56.2% when the strain was treated with 20, 50, and 100 $\mu$M oridonin, respectively (Fig. 3G).

We continued to study whether oridonin also influences intracellular c-di-GMP levels in *B. cenocepacia*. Liquid chromatography-mass spectrometry (LC-MS) analysis showed that treatment of the wild-type H111 strain with oridonin at final concentrations of 20, 50, and 100 $\mu$M caused 31.3, 58.9, and 74.9% increases in the intracellular c-di-GMP concentration, respectively (Fig. S4). However, oridonin did not bind with GtrR or RpfR (Fig. S5).

**Oridonin binds to CepR and affects its regulatory activity.** The CepIR system is conserved in *B. cepacia* complex species, in which CepR can bind to the AHL signal to form a complex and then regulate the expression of target genes by binding to the promoters (23). To explore whether oridonin can interact with the CepR protein, CepR was purified using affinity chromatography and prepared for ITC analysis (Fig. S2A). The results showed that oridonin can bind to CepR with an estimated dissociation constant ($K_D$) of 13.6 $\pm$ 0.826 $\mu$M (Fig. 4A). To further explore the relationship between oridonin and the regulatory activity of CepR, we next performed EMSA to determine whether oridonin affects the binding of CepR to the promoter DNA of its target genes.

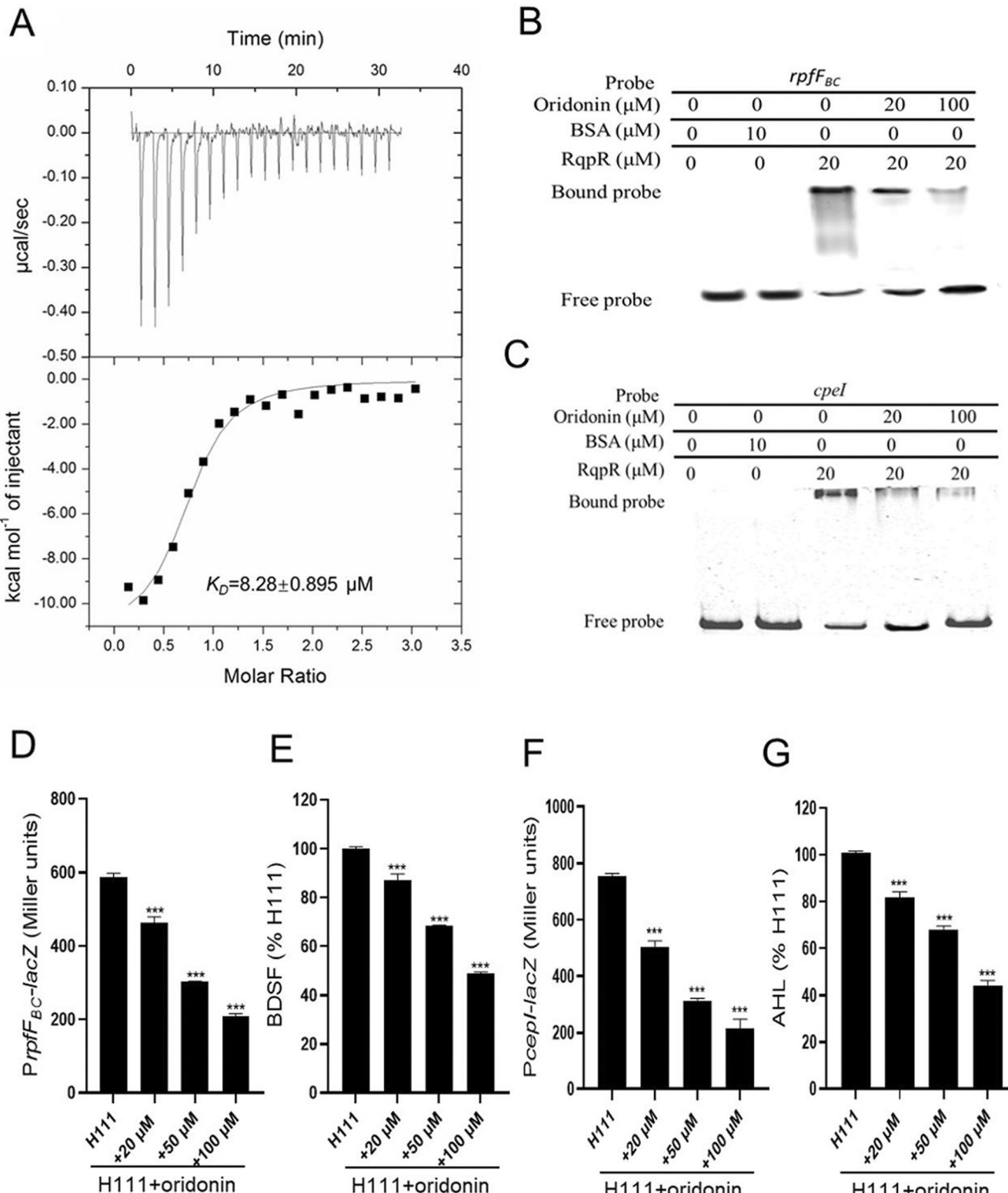

**FIG 3** Effects of oridonin on the regulatory activity of RqpR. (A) Isothermal titration calorimetry (ITC) analysis of the binding between RqpR and oridonin. (B) Electrophoretic mobility shift assay (EMSA) detection of the *in vitro* binding of RqpR to the promoter of *rpfF*$_{BC}$ in the presence of different amounts of oridonin. (C) EMSA detection of the *in vitro* binding of RqpR to the promoter of *cepI* in the presence of different amounts of oridonin. The protein was incubated with the probes in the presence of different concentrations of oridonin at room temperature for 30 min. (D) The effect of oridonin on *rpfF*$_{BC}$ gene expression was measured by assessing the β-galactosidase activity of the P*rpfF*$_{BC}$-*lacZ* transcriptional fusions. (E) Quantitative analysis of *cis*-2-dodecenoic acid (BDSF) production in *B. cenocepacia* H111 in the presence of different concentrations of oridonin (0 to 100 μM). For convenient comparison, BDSF production of *B. cenocepacia* H111 in the presence of DMSO without oridonin was arbitrarily defined as 100% and used to normalize

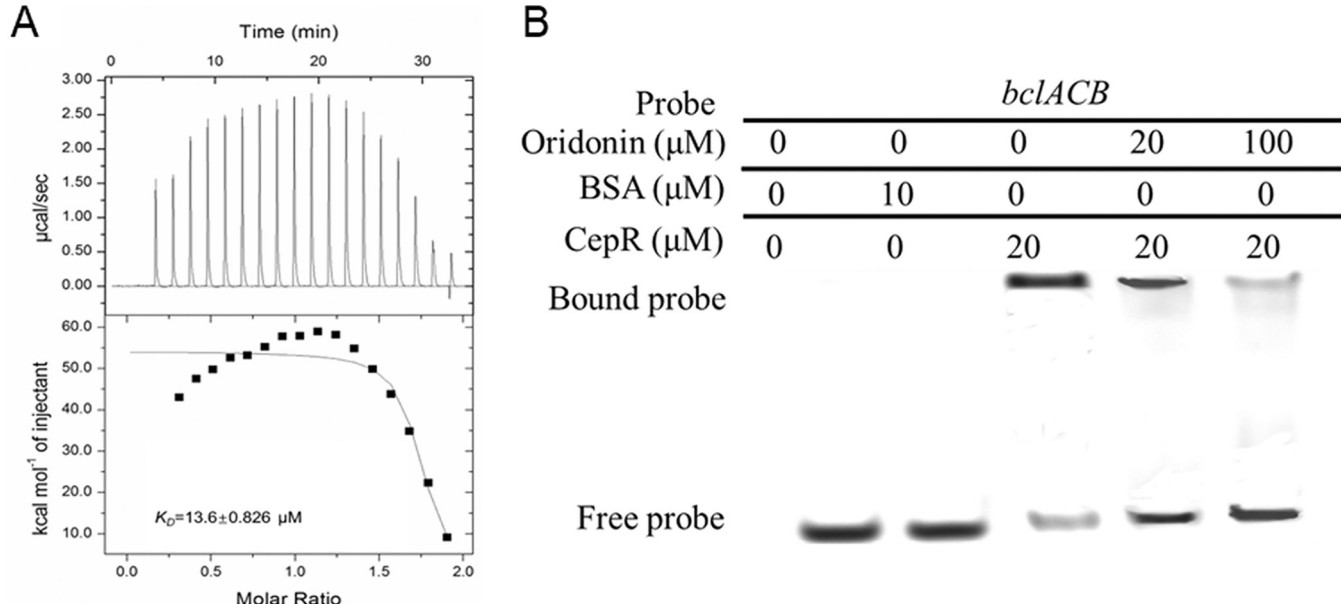

**FIG 4** Influence of oridonin on the regulatory activity of CepR. (A) ITC analysis of the binding between CepR and oridonin. (B) EMSA detection of the *in vitro* binding of CepR to the promoter of *bclACB* with the addition of different amounts of oridonin. The protein was incubated with the probe in the presence of different concentrations of oridonin at room temperature for 30 min.

As shown in Fig. 4B, the binding of CepR to the *bclACB* promoter probe was inhibited when oridonin was present in the reaction mixtures, and the amount of probe bound to the CepR protein decreased with increasing oridonin concentration.

**Oridonin impairs QS-regulated phenotypes in *B. cenocepacia*.** Exogenous addition of oridonin significantly reduced QS signal production in a dose-dependent manner but did not affect the growth of *B. cenocepacia* (Fig. 3; Fig. S3). Then, we examined the effects of different concentrations of oridonin on the phenotypes controlled by QS systems in *B. cenocepacia*. The results showed that biofilm formation, motility, and protease activity were inhibited by oridonin in a dose-dependent manner (Fig. 5). The addition of 100 $\mu$M oridonin reduced biofilm formation, motility activity, and protease activity by 68, 90, and 77%, respectively (Fig. 5). To further explore the action mechanisms of oridonin, we then tested whether BDSF and AHL signals could restore the impaired phenotypes of signal-minus mutants in the absence and presence of oridonin. It was shown that BDSF and C8-HSL increased the biofilm formation of $rpfF_{BC}$ and *cepI* mutants, respectively, in a dose-dependent manner in the absence of oridonin (Fig. S6 and S7). Exogenous addition of 50 $\mu$M BDSF and C8-HSL fully restored the impaired biofilm formation of $rpfF_{BC}$ and *cepI* mutants, respectively, in the absence of oridonin, while they exerted no any restored effects on the biofilm formation of the mutants in the presence of 100 $\mu$M oridonin (Fig. S6 and S7).

Since oridonin remarkably reduced the production of AHL and BDSF signals and impaired QS-regulated phenotypes in *B. cenocepacia* (Fig. 3 and 5), we examined whether oridonin affects the expression of genes regulated by QS systems. As shown in Fig. S8, the selected genes exhibited different expression patterns after oridonin treatment compared to the strain in the absence of oridonin. These

**FIG 3** Legend (Continued)

the signal ratios of the samples treated with oridonin. (F) The effect of oridonin on *cepI* gene expression was measured by assessing the $\beta$-galactosidase activity of the P*cepI-lacZ* transcriptional fusions. (G) Quantitative analysis of *N*-acyl homoserine lactone (AHL) production in *B. cenocepacia* H111 in the presence of different concentrations of oridonin (0 to 100 $\mu$M). For convenient comparison, AHL production of *B. cenocepacia* H111 in the presence of DMSO without oridonin was arbitrarily defined as 100% and used to normalize the signal ratios of the samples treated with oridonin. Oridonin was dissolved in DMSO, and the same volume of DMSO that was used as the solvent for the compounds was used as a control. The data are presented as the means $\pm$ standard deviations of three independent experiments. The significance of the results shown in panels D to G was determined by one-way ANOVA. *, $P < 0.05$; **, $P < 0.01$; ***, $P < 0.001$. BSA, bovine serum albumin.

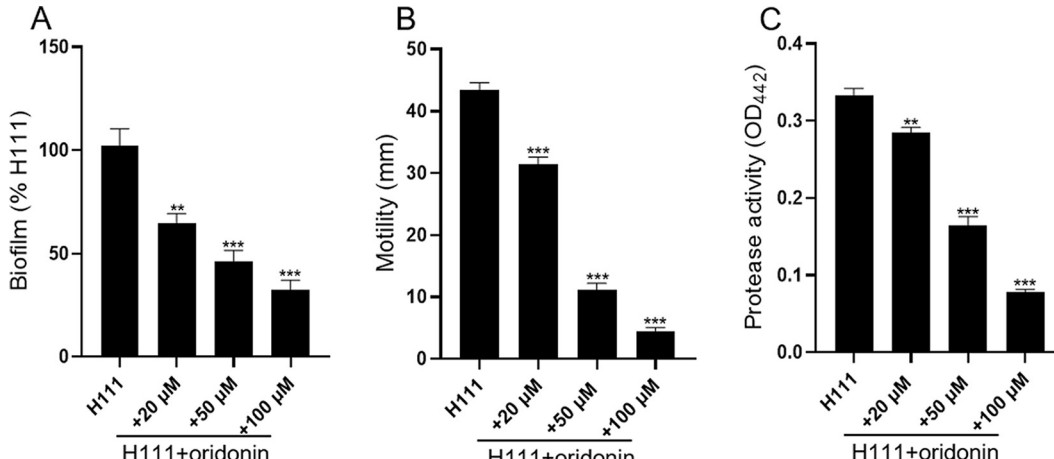

**FIG 5** Effects of oridonin on the QS-regulated phenotypes of *B. cenocepacia* H111. (A to C) Effects of oridonin on biofilm formation (A), motility (B), and protease activity (C). *B. cenocepacia* H111 was treated with different concentrations of oridonin and incubated statically at 37°C. Oridonin was dissolved in DMSO, and the same volume of DMSO used as the solvent for the compounds was used as a control. The results in the biofilm formation of *B. cenocepacia* H111 in the presence of DMSO without oridonin were defined as 100% to normalize the results of the other treatments. The data are presented as the means ± standard deviations of three independent experiments. The significance of the results was determined by one-way ANOVA. *, $P < 0.05$; **, $P < 0.01$; ***, $P < 0.001$. $OD_{442}$, optical density at 442 nm.

differentially expressed genes are involved in a range of biological functions (Table S1), suggesting that oridonin interfered with QS signaling and affected the expression of target genes in the QS systems.

**Oridonin attenuates *B. cenocepacia* virulence and suppresses the inflammation caused by *B. cenocepacia* infection.** Previous studies have shown that both the AHL and BDSF QS systems play important roles in the pathogenesis of *B. cenocepacia* (22). Since the addition of oridonin inhibited both the production of AHL and BDSF signals and cytotoxicity (Fig. 1 and 3), we used an A549 cell line infection model to examine the efficacy of different concentrations of oridonin on *B. cenocepacia* virulence. As shown in Fig. 6A, oridonin inhibited the virulence of *B. cenocepacia*, and the cytotoxicity levels were reduced to 69, 56, 23, and 14% after treatment with oridonin at final concentrations of 12.5, 25, 50, and 100 $\mu$M, respectively, while oridonin exhibited almost nontoxic effects toward A549 cells (Fig. 6B). The mortalities of mice infected with the wild-type H111 strain in the absence and presence of oridonin at a final concentration of 100 $\mu$M at 4 days postinfection were 75 and 25%, respectively (Fig. 6C).

It was previously reported that *B. cenocepacia* activates caspase-1 via NLRP3 in murine macrophages and causes pronounced inflammation (24). A previous study revealed that oridonin prevents acute inflammation and tissue damage by inhibiting the NLRP3 inflammasome (18). We next examined whether oridonin could inhibit *B. cenocepacia*-induced inflammasome activation by using the RAW 264.7 cell line. As shown in Fig. S9, *B. cenocepacia* induced the expression of inflammatory factors, such as IL-1$\beta$ and TNF-$\alpha$, and in good agreement with a previous study, the expression of both NLRP3 and caspase-1 increased, while IL-10, an anti-inflammatory factor, exhibited low expression (Fig. S9). Our results showed that exogenous addition of oridonin can effectively inhibit the expression of proinflammatory factors and promote the expression of the anti-inflammatory factor IL-10.

**Oridonin impairs QS-regulated phenotypes in many *Burkholderia* species.** Previous studies indicated that many *Burkholderia* species may employ the BDSF and AHL systems to regulate biological functions (3, 9, 20–22). Therefore, the effects of oridonin on the QS-regulated phenotypes of different *Burkholderia* species were investigated. It was found that motility activity and biofilm formation were significantly inhibited by treatment with oridonin in all the tested *Burkholderia* species, while their growth rates were unaffected (Fig. 7; Fig. S10).

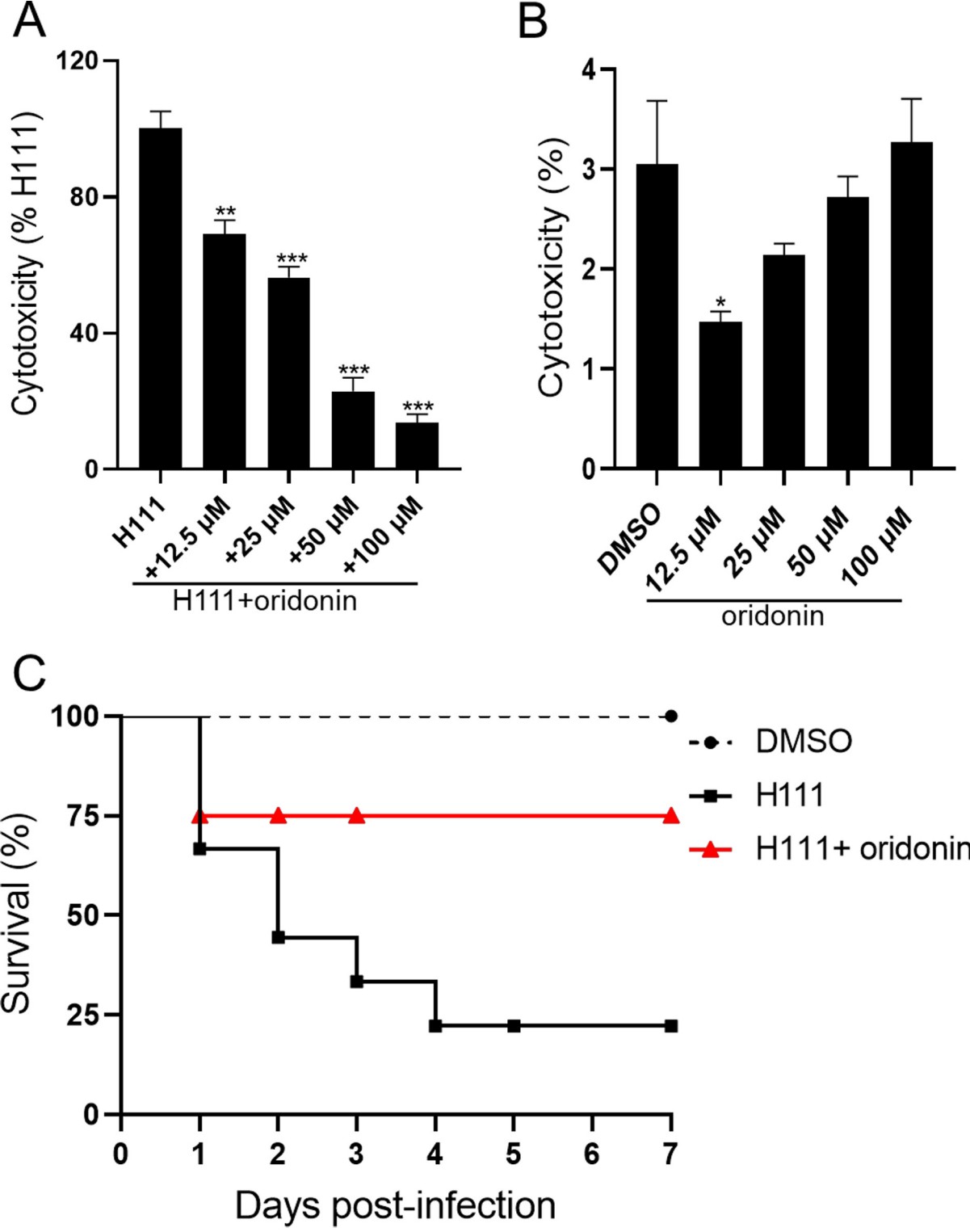

**FIG 6** Influence of oridonin on the pathogenicity of *B. cenocepacia* H111. (A) Analysis of the effect of oridonin on the cytotoxicity of *B. cenocepacia* H111. Cytotoxicity was determined by measuring LDH release. The LDH released by A549 cells after inoculation with *B. cenocepacia* H111 in the presence of DMSO without oridonin was defined as 100% to normalize the LDH release ratios of the samples treated with the different amounts of oridonin. (B) Analysis of the toxicity of oridonin to A549 cells. Oridonin was dissolved in DMSO, and the amount of DMSO used as the solvent for oridonin was used as a control. (C) Analysis of the effect of oridonin on the pathogenicity of *B. cenocepacia* H111 in a mouse infection model. Mortality was determined after BALB/c mice were infected with the *B. cenocepacia* strains in the absence or presence of oridonin over a 7-day period. The results are based on three independent experiments. The data are presented as the means ± standard deviations of three independent experiments. The significance of the results in panels A and B was determined by one-way ANOVA. *, $p < 0.05$; **, $P < 0.01$; ***, $P < 0.001$.

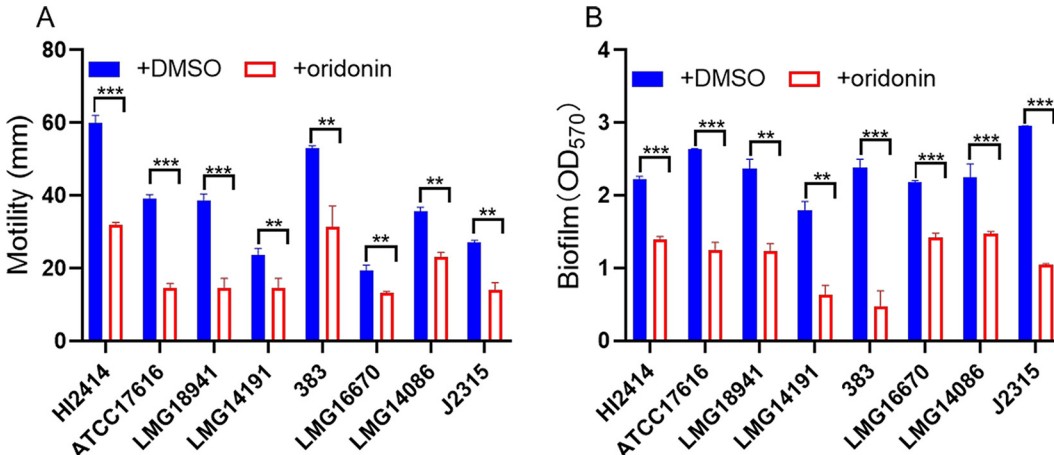

**FIG 7** Quantitative analyses of motility (A) and biofilm formation (B) of different *Burkholderia* species in the absence or presence of oridonin (100 $\mu$M). HI2414, ATCC 17616, LMG 18941, LMG 14191, AU10541, LMG 16670, LMG 14086, and J2315 represent *B. cenocepacia* HI2414, *B. multivorans* ATCC 17616, *B. dolosa* LMG 18941, *B. prymocina* LMG 14191, *B. cepacia* 383, *B. anthina* LMG 16670, *B. stabilis* LMG 14086, and *B. cenocepacia* J2315, respectively. The cells were treated with oridonin at a final concentration of 100 $\mu$M and incubated statically at 37°C. Oridonin was dissolved in DMSO, and the same volume of DMSO that was used as the solvent for the compounds was used as a control. The results are based on three independent experiments. The data are presented as the means $\pm$ standard deviations of three independent experiments. The significance of the results in panels A and B was determined by one-way ANOVA. *, $P < 0.05$; **, $P < 0.01$; ***, $P < 0.001$.

## DISCUSSION

Oridonin, as the famous ent-kaurane diterpenoid isolated from *I. rubescens*, has been confirmed to exhibit many kinds of biological properties, including anticancer, anti-inflammation, and antioxidation activities (25–28). It was previously reported that oridonin can inhibit the production of reactive oxygen species (ROS) to induce apoptosis, ultimately exerting antitumor effects (29), and oridonin can also alleviate the lipopolysaccharide (LPS)-induced inflammatory response via the NF-$\kappa$B pathway (30, 31). In addition, oridonin has been shown to inhibit the growth of several bacteria, including *Salmonella typhi*, *Streptococcus pneumoniae*, *Shigella castellani*, and *Staphylococcus aureus* (32, 33). Although the antibacterial activity of oridonin was previously identified, the underlying mechanisms and direct targets remain unknown. In this study, we found that oridonin can directly bind to RqpR and CepR, influence QS, and affect the biological functions of *B. cenocepacia* (Fig. 3 to 5; Fig. S8). Intriguingly, oridonin did not affect the growth of *B. cenocepacia* cells (Fig. S3), suggesting its excellent antivirulence activity. Our results demonstrate that oridonin has antimicrobial efficacy against *B. cenocepacia* by interfering with virulence, which is a mechanism that is different from that of traditional antibiotics.

*B. cenocepacia* is an important opportunistic human pathogen that causes substantial damage to pulmonary function in patients with the genetic disease cystic fibrosis (CF) and is a critical nosocomial pathogen causing bacteremia and urinary tract infections (1, 34). Previous studies have demonstrated that *B. cenocepacia* infection is associated with a quick decline in lung function and increased mortality (35). In addition, *B. cenocepacia* has multiple mechanisms to respond to antibiotics. Previous studies have found that *B. cenocepacia* produces 4-amino-4-deoxy-L-arabinose to reduce its polymyxin susceptibility (36). In addition, several efflux pumps are encoded in *B. cenocepacia*; for example, NorM has been shown to play a role in polymyxin resistance (37); resistance nodulation cell division (RND) efflux pump encoding is able to confer resistance to clinically relevant antibiotics, such as aminoglycosides, chloramphenicol, fluoroquinolones, and tetracyclines (38, 39); and BcrA is involved in resistance to tetracycline and quinolones (40). Such resistance to antibiotics complicates disease management and treatment strategies (41). To solve this problem, it is urgent to find new strategies to treat *B. cenocepacia*.

The multidrug-resistant *B. cenocepacia* that is responsible for infections in patients with CF has caused increasing concern worldwide due to the complexity and difficulty of its treatment (41). Following the "golden age" of antibiotic discovery, the use and misuse of antibiotics has resulted in the emergence of antimicrobial-resistant bacteria (42). Now, new indirect therapeutic approaches have been developed to avoid the emergence of drug-resistant bacteria, including the inhibition of biofilm formation, motility, virulence, and QS signaling (13, 43). In particular, the development of QS inhibitors as novel antimicrobial drugs has been of great interest over the past 20 years (44–48). Various strategies have been explored to control QS signaling, including suppressing signal synthesis, interfering with signal sensing, and promoting signal degradation (49). In our previous study, the structural BDSF analog *cis*-14-methylpentadec-2-enoic acid, as a QS signal inhibitor, showed strong interference with BDSF signaling and virulence but did not inhibit the growth rate of *B. cenocepacia* cells (21). In this study, oridonin was identified to interfere with the QS of *B. cenocepacia* by directly binding to RqpR and CepR (Fig. 3 and 4) but did not inhibit the growth rate to avoid the spread of resistant bacteria (Fig. S3). In addition, we found that oridonin could inhibit biofilm formation, motility, and protease activity and attenuated the virulence of *B. cenocepacia* while showing very low toxicity to human cells (Fig. 6). It was previously reported that *B. cenocepacia* promotes the outbreak of inflammatory factors via NLRP3 to cause damage to patients (24). Our findings revealed that oridonin could inhibit inflammation caused by *B. cenocepacia* (Fig. S9). In summary, our study indicated that oridonin possesses significant value and potential development prospects as a QS inhibitor.

## MATERIALS AND METHODS

**Ethics statement.** This study was approved by the ethics committee of School of Pharmaceutical Sciences (Shenzhen), Sun Yat-sen University under approval no. SYSU-20200404, and all participants gave informed consent.

**Bacterial strains and growth conditions.** All the strains used in this study are listed in Table 1. The *B. cepacia* complex, *B. cenocepacia* H111, $rpfF_{BC}$ mutant, *cepI* mutant, and *Escherichia coli* strains were grown at 37°C in LB medium (5 g yeast extract, 10 g tryptone, and 10 g/liter NaCl; solid medium also contained 15 g/liter agar). The following antibiotics were used to supplement the media when necessary: 100 $\mu$g/mL ampicillin, 100 $\mu$g/mL kanamycin, and 20 $\mu$g/mL tetracycline. The chromogenic substrate X-Gal (5-bromo-4-chloro-3-indolyl $\beta$-D-galactopyranoside) was used at 40 $\mu$g/mL. Oridonin (Yuanye Bio-Technology, Shanghai, China; high-pressure liquid chromatography [HPLC] $\geq$ 99%) was dissolved in dimethyl sulfoxide (DMSO) to a final concentration of 100 mM, and this solution was added to the medium in the experiments. In the protease activity experiment, NYG medium (3 g yeast extract, 5 g peptone, and 20 g/liter glycerin) was used to culture *B. cenocepacia* H111. In the biofilm formation assay, minimal medium (2 g glycerin, 2 g mannitol, 10.5 g $K_2HPO_4$, 4.5 g $KH_2PO_4$, 2 g $(NH_4)_2SO_4$, 0.2 g $MgSO_4·7H_2O$, 0.005 g $FeSO_4$, 0.01 g $CaCl_2$, and 0.002 g/liter $MnCl_2$) was used to culture *B. cenocepacia* H111. Bacterial growth was monitored spectrophotometrically by measuring the optical density at 600 nm.

**$\beta$-Galactosidase activity assay.** *bclACB*, *cepI*, and $rpfF_{BC}$ reporter strains in the absence or presence of different concentrations of oridonin were cultured overnight in LB medium supplemented with ampicillin and tetracycline at 37°C. Then, the cells were harvested, and $\beta$-galactosidase activity was measured following previously described methods (3).

**Swarming motility assay.** *B. cepacia* complex motility was determined on semisolid agar (8 g tryptone, 5 g glucose, and 3 g/liter agarose). The bacteria were inoculated into the centers of the plates, which contained different concentrations of oridonin. The plates were incubated at 30°C for 18 h before the diameters of the colonies were measured (9).

**Biofilm formation assay.** The biofilm formation assay was performed as previously described with minor modifications (50). Overnight cultures of bacterial cells were diluted to an optical density at 600 nm ($OD_{600}$) of 0.01 by using LB medium, added to 96-well polystyrene plates supplemented with different concentrations of oridonin or QS signals, and incubated at 37°C for 24 h. Then, the culture medium was poured out, and the wells were stained with crystal violet for 15 min followed by washing three times with water before the addition of 95% ethanol. Biofilm formation was quantified by measuring the absorbance at 570 nm.

**Protease activity assay.** Protease activity assays were performed as previously described with minor modifications (51). *B. cenocepacia* H111 was cultured overnight in 10 mL of NYG medium at 37°C ($OD_{600} = 4.0$) in the absence or presence of oridonin at final concentrations of 20, 50, and 100 $\mu$M. The cultures were then centrifuged at 13,000 rpm for 10 min. The collected supernatants were mixed with azocasein solution (5 g azocasein and 7.882 g/liter Tris, pH 8.0) and incubated in a water bath at a constant temperature of 30°C for 60 min. Then, 400 $\mu$L of 10% trichloroacetic acid (TCA; 100 g/liter) was added to terminate the reaction. After centrifugation at 13,000 rpm for 2 min, the supernatants were mixed with 700 $\mu$L of 525 mM NaOH solution. Protease activity was quantified by measuring the absorbance at 442 nm.

**Quantification of BDSF and c-di-GMP.** *B. cenocepacia* H111 with and without oridonin at final concentrations of 20, 50, and 100 $\mu$M were cultured overnight in LB broth at 37°C ($OD_{600} = 3.0$). The cultures

**TABLE 1** Bacterial strains and plasmids used in this study[a]

| Strain or plasmid | Phenotypes and/or characteristics | Reference or source |
|---|---|---|
| *Burkholderia* spp. | | |
| *B. cenocepacia* H111 | Wild-type strain, genomovars III of the *B. cepacia* complex | 50 |
| $\Delta rpfF_{BC}$ | BDSF-minus mutant derived from H111 with $rpfF_{BC}$ being deleted | 6 |
| $\Delta cepI$ | Deletion mutant with *cepI* being deleted | 22 |
| H111(P*bclACB-lacZ*) | H111 harboring the reporter construct P*bclACB-lacZ* | 3 |
| H111 (P*cepI*-lacZ) | H111 harboring the reporter construct P*cepI*-lacZ | 3 |
| H111(P*rpfF_{BC}-lacZ*) | H111 harboring the reporter construct P*rpfF_{BC}-lacZ* | 3 |
| *B. cenocepacia* HI2414 | Isolated from agricultural soil, USA | BccM |
| *B. multivorans* ATCC 17616 | Soil, USA | E. Mahenthiralingam's laboratory |
| *B. dolosa* LMG 18941 | Cystic fibrosis isolate, USA | BccM |
| *B. prymocina* LMG 14191 | Soil, Japan | BccM |
| *B. cepacia* 383 | Soil, Trinidad | E. Mahenthiralingam's laboratory |
| *B. anthina* LMG 16670 | Rhizosphere, UK | BccM |
| *B. stabilis* LMG 14086 | Respirator, UK | BccM |
| *B. cenocepacia* J2315 | Cystic fibrosis isolate, UK | ATCC |
| | | |
| *E. coli* | | |
| DH5$\alpha$ | *supE44 lacU169 (80lacZM15) hsdR17 recA1 endA1 gyrA96 thi-1 relA1 pir* | Laboratory collection |
| BL21 | F-*ompT hsdS* ($r_{B-}m_{B-}$) *dcm*$^+$ Tet$^r$ *gal* (DE3) *endA* | Stratagene |
| | | |
| Plasmid | | |
| pME2-*lacZ* | Transcriptional level report vector, Tet$^r$ | Laboratory collection |
| P*bclACB-lacZ* | pME2-*lacZ* containing promoter of *bclACB* | 3 |
| P*cepI*-lacZ | pME2-*lacZ* containing promoter of *cepI* | 3 |
| P*rpfF_{BC}-lacZ* | pME2-*lacZ* containing promoter of $rpfF_{BC}$ | 3 |
| pDBHT2 | Expression vector, Kan$^r$ | Laboratory collection |
| Pet28a | Expression vector, Kan$^r$ | Novagen |
| pDBHT2-*cepR* | pDBHT2 containing *cepR* | 20 |
| pET28a-*gtrR* | pET28a containing *gtrR* | 3 |
| pET28a-*rpfR* | pET28a containing *rpfR* | 3 |
| pDBHT2-*rpqR* | pDBHT2 containing *rpqR* | This study |

[a]Kan$^r$, resistance to kanamycin; Tet$^r$, resistance to tetracycline; Amp$^r$, resistance to ampicillin; Gm$^r$ resistance to gentamicin; BccM, Belgian Coordinated Collections of Microorganisms; BDSF, *cis*-2-dodecenoic acid.

were centrifuged at 5,000 rpm for 10 min, and the supernatants were mixed with an equal volume of ethyl acetate. The ethyl acetate fractions were collected, evaporated to dryness, and dissolved in 1 mL of methanol. BDSF signals were measured by liquid chromatography-mass spectrometry (LC-MS) (52).

*B. cenocepacia* H111 were cultured overnight in LB broth at 37°C with and without oridonin at final concentrations of 20, 50, and 100 $\mu$M (OD$_{600}$ = 3.0). Formaldehyde (final concentration, 0.18%) was added to block the degradation of c-di-GMP. The cultures were centrifuged at 5,000 rpm for 10 min. The cell pellets were washed with 40 mL of phosphate-buffered saline (pH 7.0) containing 0.18% formaldehyde and centrifuged at 8,000 rpm for 10 min at 4°C. Next, the cell pellets were dissolved in water, boiled for 10 min, and cooled on ice for 10 min. The nucleotides were extracted using 65% ethanol. The supernatants were retained, and the extractions were repeated. The supernatants were concentrated and lyophilized, and the pellets were dissolved in 1 mL of H$_2$O; c-di-GMP levels were measured by LC-MS (6).

**Quantification of AHL signals.** Quantification of AHL signals was performed using the $\beta$-galactosidase assay with the aid of the AHL reporter strain CF11, as described previously (53). Briefly, the reporter strain CF11 was cultured in minimal medium at 28°C for 12 h with the addition of different concentrations of oridonin (final concentrations, 20, 50, and 100 $\mu$M). The cultures were inoculated into the same medium supplemented with extracts containing AHL signals. After bacterial cells were harvested, $\beta$-galactosidase activity was assayed as described previously (3).

**Protein expression and purification.** Affinity purification of the HIS-CepR, HIS-GtrR, HIS-RpfR, and HIS-RqpR fusion proteins was performed following methods described previously, while the strain with pDBHT2-CepR was cultured with the addition of C8-HSL (50 nM) (3, 9, 20). Fusion protein cleavage with TEV protease (Beyotime, Shanghai, China) was conducted at 4°C overnight. The purified proteins were eluted and verified by SDS-PAGE.

**Isothermal titration calorimetry analysis.** Isothermal titration calorimetry measurements were performed using an ITC-200 microcalorimeter following the manufacturer's protocol (MicroCal, Northampton, MA) (54). In brief, titrations began with one injection of 0.2 $\mu$L of oridonin (200 $\mu$M) solution into the sample cell containing 350 $\mu$L of the protein solution (20 $\mu$M). The volume of the oridonin injection was changed to 2 $\mu$L in the subsequent 19 injections. The heat changes accompanying the injections were recorded. The titration experiment was repeated at least three times, and the data were calibrated with the final injections and fitted to the one-site model to determine the binding constant ($K_D$) using MicroCal Origin version 7 software.

**TABLE 2** PCR primers used in this study[a]

| Primer | Sequence (5′ to 3′) |
|---|---|
| **For EMSA** | |
| EMSA-*bclACB*-F | GATGTCGGTCCTCGGTCT |
| EMSA-*bclACB*-R | CGAACATGAATAGGGCCT |
| EMSA-*rpfF$_{BC}$*-F | GGTATGTCCTCGTGAGATGTGGT |
| EMSA-*rpfF$_{BC}$*-R | GTCGAAGCTCTCCGCGCG |
| | |
| **For recombinant protein** | |
| *cepR*-HIS-F | CGGGATCCATGGAACTGCGCTGGCAG |
| *cepR*-HIS-R | CGGAATTCTCAGGGTGCTTCGATGAG |
| *rqpR*-HIS-F | CGGGATCCATGAGCCTGAACATCCTGCTCG |
| *rqpR*-HIS-R | CGGAATTCTCAGGCGCCGGCCGTGGG |
| | |
| **For RT-qPCR** | |
| *recA*-F | GTACGATCAAGCGCACGAAC |
| *recA*-R | GATCCGGCGGATATCGAGAC |
| *BCAL0124*-F | ACCTGTCGTACCTCCTCCTC |
| *BCAL0124*-R | CGTGATCATCGAAGCGGAAG |
| *BCAL0833*-F | TAGTCGTCACGTATTCGCCG |
| *BCAL0833*-R | CTTCTCGATGCATTGCTGGC |
| *BCAM0184*-F | CAACCCTTTACCCACGACGA |
| *BCAM0184*-R | CGTATTGCGGCAGTTTCTCG |
| *BCAM0193*-F | GCACGACTACCACGAGGAAG |
| *BCAM0193*-R | GAAGTAGCTGCCTTCCCGAT |
| *BCAM1010*-F | TGTCGGGCATCATCGAGAAG |
| *BCAM1010*-R | GCTTGCGCAGATGATCGAAG |
| *BCAM1745*-F | CCGACATCATCCTGCTCGAA |
| *BCAM1745*-R | TGGCCGTCATGTTCAGGTAC |
| *BCAM1870*-F | AGTTCGATCGCGACGATACC |
| *BCAM1870*-R | AGCGACTTCAGCAGATACGG |
| *BCAM1871*-F | CTCGAACGACAGGTTGACGA |
| *BCAM1871*-R | GTATTTGCTGCGCATCTCCG |
| *BCAM2140*-F | AATTCTCGACGAAGCTCGCA |
| *BCAM2140*-R | GATGTCTTTCACGATGCCGC |
| *BCAM2143*-F | GACGATCCAGGTCGATGGTC |
| *BCAM2143*-R | GTATCCACCACGATCCCCAC |
| *BCAM2227*-F | ACAGGAAGGCTTGTCGGAAG |
| *BCAM2227*-R | CGTCCCAGTTGTAGACCCAG |
| *BCAM2307*-F | GATGGACAAGGCGTTCCTGA |
| *BCAM2307*-R | GTGCAGCTCTTGTTGTACGC |
| *BCAS0292*-F | GTCTGGTGTTCGTTGCGATG |
| *BCAS0292*-R | CAAAGAGCCGGTTGTCGTTG |
| *BCAL0524*-F | CAGATGGTGCTCAAGGAAGT |
| *BCAL0524*-R | GACATGTTCGCGAGGAACT |
| *BCAM0854*-F | GGGACGATGGCGATTTCTT |
| *BCAM0854*-R | GGTTCCATCACCGCATAGTC |
| *gapdh*-F | AACGGATTTGGTCGTATTG |
| *gapdh*-R | GCTCCTGGAAGATGGTGAT |
| *NLRP3*-F | AAAGCCAAGAATCCACAGTGTAAC |
| *NLRP3*-R | TTGCCTCGCAGGTAAAGGT |
| *caspase1*-F | AGGCATGACAATGCTGCTACAA |
| *caspase1*-R | TGTGCAAATGCCTCCAGCTC |
| *ASC*-F | GGATGCTCTGTACGGGAAGG |
| *ASC*-R | CGCATCTTGCTTGGGTTG |
| *IL-1β*-F | ACAGTGGCAATGGAGGATGAC |
| *IL-1β*-R | AGGTGCATCGTGCACATAAG |
| *IL-18*-F | GGCCTCTATTTGAAGATATGACTGATT |
| *IL-18*-R | CCATACCTCTAGGCTGGCTATCTTT |
| *TNF-α*-F | TGCTCCTCACCCACACCAT |
| *TNF-α*-R | GCCCAGACTCGGCAAAGTC |
| *IL-10*-F | CGAGATGCCTTCAGCAGAGTG |
| *IL-10*-R | TCATCTCAGAACAAGGCTTGGC |

[a]Restriction enzyme sites are underlined. EMSA, electrophoretic mobility shift assay; F, forward; R, reverse; RT, reverse transcription.

**Electrophoretic mobility shift assay (EMSA).** The DNA probes used for the EMSA were harvested by PCR amplification using the primer pairs listed in Table 2. The purified PCR products of the *bclACB* promoters were 3′-end-labeled with biotin according to the manufacturer's instructions (Thermo Fisher, Waltham, MA). The biotin-labeled probes and proteins were prepared for the DNA-protein binding reactions following the manufacturer's instructions (Thermo Fisher, Waltham, MA). A 5% polyacrylamide gel was used to separate the DNA-protein complexes from the unbound probes following the methods described previously (3). After UV cross-linking, the biotin-labeled probes were detected in the membrane with different mobilities between the bound probes and unbound probes.

**Real-time quantitative reverse transcription-PCR assay.** *B. cenocepacia* H111 with and without oridonin at a final concentration of 100 $\mu$M were cultured at 37°C in LB broth to an $OD_{600}$ of 1.0 and then harvested. The adherent cells together with cells in suspension were collected for RNA extraction for reverse transcription. RNA was isolated using an Eastep Super total RNA extraction kit (Promega, Madison, WI). cDNA synthesis and reverse transcription-quantitative PCR (RT-qPCR) analysis were performed with ChamQ Universal SYBR qPCR Master Mix (Vazyme, Nanjing, China) according to the manufacturer's instructions in a 7300Plus quantitative real-time PCR system. As a control, the expression of the *recA* gene was analyzed by RT-qPCR. The relative expression levels of the target genes were calculated using the comparative CT ($2^{-\Delta\Delta CT}$) method (55).

RAW 264.7 cells were adjusted to a concentration of $5 \times 10^6$ cells/mL, seeded into culture flasks, and cultured at 37°C under 5% $CO_2$. RAW264.7 cells were infected with bacterial cells at $10^9$ CFU/mL in the presence or absence of oridonin for 8 h. Oridonin was added to the cells at a final concentration of 1 or 5 $\mu$M. RNA was isolated using an Eastep Super total RNA extraction kit (Promega, Madison, WI). cDNA synthesis and quantitative RT-qPCR analysis were performed with ChamQ Universal SYBR qPCR Master Mix (Vazyme, Nanjing, China) according to the manufacturer's instructions in a 7300Plus quantitative real-time PCR system. As a control, the expression of the *gapdh* gene was analyzed by RT-qPCR. The relative expression levels of the target genes were calculated using the comparative CT ($2^{-\Delta\Delta CT}$) method (55).

**Cytotoxicity assays.** Cytotoxicity assays were performed according to previously described methods (21). In brief, *B. cenocepacia* H111 with the addition of different concentrations of oridonin was cultured in LB medium at 37°C overnight, centrifuged, and resuspended in Dulbecco's modified Eagle's medium (DMEM; 1% fetal bovine serum [FBS]) to an $OD_{600}$ of 1.0. A549 cells were infected with bacterial cells at $10^9$ CFU/mL for 8 h. The amount of LDH released was determined with a cytoTOX96 kit (Promega, Madison, WI). The results of the cytotoxicity assay were quantified by measuring the absorbance at 490 nm, and the cytotoxicity was calculated relative to that of an uninfected control.

**Mouse infections.** The animal infection experiments were based on a published study with minor modifications (9). This experiment was performed using BALB/c mice and conducted according to the National Institutes of Health Guide for the Care and Use of Laboratory Animals (NIH publication no. 8023, revised 1978). Male mice (6 to 8 weeks old,18 to 20 g) were randomly divided into groups. The mice were infected with 400 $\mu$L of inoculum containing bacterial cells ($OD_{600} = 0.8$) in the absence or presence of oridonin at a final concentration of 100 $\mu$M via intraperitoneal injection. Phosphate-buffered saline (PBS) solution was also injected into the mice as a blank control. The death rates were determined over the first 7 days postinfection.

**Statistical analysis.** Statistical analyses were performed with GraphPad Prism 8. The data are presented as the means $\pm$ standard deviations of three independent experiments. Statistical significance is indicated as follows: *, $P < 0.05$; **, $P < 0.01$; ***, $P < 0.001$ (one-way analysis of variance [ANOVA] or two-way ANOVA). All results were calculated from the means of at least three replicates.

## SUPPLEMENTAL MATERIAL

Supplemental material is available online only.

**SUPPLEMENTAL FILE 1**, PDF file, 0.8 MB.

## ACKNOWLEDGMENTS

This work was financially supported by grant 2021YFA0717003 of the National Key Research and Development Program of China and grant JCYJ20200109142416497 from the Science, Technology and Innovation Commission of Shenzhen Municipality.

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
