## [Reviewer comments · Microbiology Spectrum]

Microbiology Spectrum

Oridonin attenuates *Burkholderia cenocepacia* virulence by suppressing quorum sensing signaling

Xia Li, Kai Wang, Gerun Wang, Binbin Cui, Shihao Song, Xiuyun Sun, and Yinyue Deng

Corresponding Author(s): Yinyue Deng, Sun Yat-sen University

Review Timeline:

Submission Date:	May 13, 2022
Editorial Decision:	June 9, 2022
Revision Received:	June 25, 2022
Accepted:	July 4, 2022

Editor: Giordano Rampioni

Reviewer(s): Disclosure of reviewer identity is with reference to reviewer comments included in decision letter(s). The following individuals involved in review of your submission have agreed to reveal their identity: Eunhye Goo (Reviewer #2)

Transaction Report:

DOI: <https://doi.org/10.1128/spectrum.01787-22>

June 9, 2022

Prof. Yinyue Deng
Sun Yat-sen University
Guangzhou 510642
China

Re: Spectrum01787-22 (Oridonin attenuates *Burkholderia cenocepacia* virulence by suppressing quorum sensing signaling)

Dear Prof. Yinyue Deng:

Thank you for submitting your manuscript to Microbiology Spectrum. Your manuscript has been evaluated by two Reviewers with expertise in the area addressed in your study and it was the consensus view of these Reviewers that your paper reports interesting data. However, both Reviewers recommend modifications before manuscript acceptance, including key control experiments to evaluate possible oridonin cytotoxicity, and to support and elucidate its specific mechanism of action. I will be glad to consider for publication in Microbiology Spectrum a revised version of your manuscript addressing all the comments raised by the Reviewers.

Link Not Available

Sincerely,

Giordano Rampioni

Journals Department
Reviewer comments:

Reviewer #1 (Comments for the Author):

in this work, Li et al identify oridonin among more than 1,000 natural compounds as an antivirulence and anti-quorum sensing molecule in *Burkholderia cenocepacia*. They claim that oridonin could be used for treating *Burkholderia* infections.

Major comments

There is an antivirulence screen with more than 1,000 natural products that could be made available

Fig. 1 and 2 show several compounds that seem to have similar activity to that of oridonin. However, the choice of oridonin for follow-up studies is not justified.

Line 136-139. There is a knowledge gap between a compound's first report of activity and the identification of its putative binding target. It is not clear what knowledge or findings led the authors to hypothesize that RqpR could be a binding target of oridonin.

Fig. 3B. The EMSA results are not very clear. The free probe does not decrease with the addition of oridonin.

The literature reports several effects of Oridonin seems on many cellular process. It seems that many of these claims could be due to unspecific binding, protein aggregation, protein precipitation. Have the authors considered this possibility?

In order to show activity, most of the assays are performed at high concentrations(100uM). There is plenty of data on the toxic effects of oridonin. Have the authors look at the toxic effects at these concentrations?

Figure 6A and 6B. It is not clear how the authors calculated the percentages. The y-axis of figure 6B is labeled % WT. It is not clear what that means.

The idea of targeting quorum sensing as an antivirulent target is not new. However, it did not so far render any promising results. The authors should exert caution on suggesting this approach as promising. Quorum sensing in Burkholderia is complex and not conserved in the different species. In addition, quorum sensing may not be relevant at chronic stages of infection.

Reviewer #2 (Comments for the Author):

This study showed the inhibitory efficacy of oridonin against two types of quorum sensing in the pathogenic Burkholderia cenocapacia. The various experimental data support the conclusions, and the manuscript is written in standard English and easy to comprehend. However,

1. Lack of description of the rationale for choosing oridonin as a QS inhibitor among all the other candidates.
2. The concentration of 100 μ M, which showed effectiveness as a QS inhibitor, is pretty much high. Is it economical for practical use?
3. There is no discussion about the mode of action of oridonin as a QS inhibitor. Since the chemical structures of oridonin, BDSF and C8-HSL are different, it is less likely that the oridonin is a competitive inhibitor. It can easily get the evidence through a simple experiment to determine whether the oridonin is a non-competitive inhibitor. Please test whether increasing the concentration of C8-HSL does not affect biofilm, motility, or protease activity in the cepl mutant in the absence or presence of oridonin at the IC50 value.

The following are minor points that are needed to improve the manuscript.

- 1) Lines 97, 215: "Burkholderia" should be italic.
- 2) Lines 111-113: data not shown?
- 3) Lines 118-121: The levels of cytotoxicity of bilirubin look similar to its theaflavin 3'3-digallate. Please clarify the percentage of cytotoxicity as the baseline for choosing the compounds.
- 4) Lines 128-133: Based on your data (Fig. 2A), β -hydroxyisovalerylshikonin is not an effective inhibitor against rpfF gene expression. Same as the theaflavin 3'3-digallate, protopseuodohypericin, ginsenoside Rk1, theaflavin-3-digallate, and α -boswellic acid. Please consider re-select the compounds which show the significant differences.
- 5) Lines 200-201: Please explain the reason in the Discussion.
- 6) Methods: If you used a commercial product of oridonin, please mention the company of the product.
- 7) Figure 4B and lines 173-175: The intensities of the band of the bound probe are pretty much the same between 20 μ M and 10 μ M of oridonin. It seems that the amount of the bclACB promoter probe bound to the CepR did not decrease dependent on increasing oridonin concentration.
- 8) Method: In general, QS signal receptor proteins are insoluble when overexpressed without the cognate signals. Please describe the details of the purification process of signal receptor proteins, and indicate whether the QS signals were added to the EMSA reaction.

Staff Comments:

Preparing Revision Guidelines

To submit your modified manuscript, log onto the eJP submission site at <https://spectrum.msubmit.net/cgi-bin/main.plex>. Go to Author Tasks and click the appropriate manuscript title to begin the revision process. The information that you entered when you first submitted the paper will be displayed. Please update the information as necessary. Here are a few examples of required

updates that authors must address:

Please return the manuscript within 60 days; if you cannot complete the modification within this time period, please contact me. If you do not wish to modify the manuscript and prefer to submit it to another journal, please notify me of your decision immediately so that the manuscript may be formally withdrawn from consideration by Microbiology Spectrum.

Comments and Suggestions for the Author:

This study showed the inhibitory efficacy of oridonin against two types of quorum sensing in the pathogenic *Burkholderia cenocapacia*. The various experimental data support the conclusions, and the manuscript is written in standard English and easy to comprehend. However,

1. Lack of description of the rationale for choosing oridonin as a QS inhibitor among all the other candidates.
2. The concentration of 100 μM , which showed effectiveness as a QS inhibitor, is pretty much high. Is it economical for practical use?
3. There is no discussion about the mode of action of oridonin as a QS inhibitor. Since the chemical structures of oridonin, BDSF and C8-HSL are different, it is less likely that the oridonin is a competitive inhibitor. It can easily get the evidence through a simple experiment to determine whether the oridonin is a non-competitive inhibitor. Please test whether increasing the concentration of C8-HSL does not affect biofilm, motility, or protease activity in the *cepI* mutant in the absence or presence of oridonin at the IC_{50} value.

The following are minor points that are needed to improve the manuscript.

- 1) Lines 97, 215: "*Burkholderia*" should be italic.
- 2) Lines 111-113: data not shown?
- 3) Lines 118-121: The levels of cytotoxicity of bilirubin look similar to its theaflavin 3'3-digallate. Please clarify the percentage of cytotoxicity as the baseline for choosing the compounds.
- 4) Lines 128-133: Based on your data (Fig. 2A), β -hydroxyisovalerylshikonic acid is not an effective inhibitor against *rpfF* gene expression. Same as the theaflavin 3'3-digallate, protopseuodohypericin, ginsenoside Rk1, theaflavin-3-digallate, and α -boswellic acid. Please consider re-select the compounds which show the significant differences.
- 5) Lines 200-201: Please explain the reason in the Discussion.
- 6) Methods: If you used a commercial product of oridonin, please mention the company of the product.
- 7) Figure 4B and lines 173-175: The intensities of the band of the bound probe are pretty much the same between 20 μM and 10 μM of oridonin. It seems that the amount of the *bclACB* promoter probe bound to the CepR did not decrease dependent on increasing oridonin concentration.
- 8) Method: In general, QS signal receptor proteins are insoluble when overexpressed without the cognate signals. Please describe the details of the purification process of signal receptor proteins, and indicate whether the QS signals were added to the EMSA reaction.

Point-to-point response to reviewers' suggestions

Reviewer comments:

Reviewer #1 (Comments for the Author):

in this work, Li et al identify oridonin among more than 1,000 natural compounds as an antivirulence and anti-quorum sensing molecule in *Burkholderia cenocepacia*. They claim that oridonin could be used for treating *Burkholderia* infections.

Major comments

There is an antivirulence screen with more than 1,000 natural products that could be made available

Fig. 1 and 2 show several compounds that seem to have similar activity to that of oridonin. However, the choice of oridonin for follow-up studies is not justified.

Response: Good suggestion. Among these thirteen active candidate compounds, theaflavin-3,3'-digallate, thoningianin A, oridonin and acetyl-alpha-boswellic acid exhibited good inhibitory activity on both *rpfF_{BC}* and *cepl* gene expression. However, we have only identified the direct targets of oridonin. Isothermal Titration Calorimetry (ITC) analysis showed that only oridonin bound to RqpR with an estimated dissociation constant (K_D) of $8.28 \pm 0.895 \mu\text{M}$ (Fig.3, Fig. S2). So, we chose oridonin for further investigation in this study.

Line 136-139. There is a knowledge gap between a compound's first report of activity and the identification of its putative binding target. It is not clear what knowledge or findings led the authors to hypothesize that RqpR could be a binding target of oridonin.

Response: Good suggestion. Our previous study showed that the novel two-component system RqpSR directly controls the BDSF and AHL QS systems in *B. cenocepacia* (Cui et al., *Molecular Microbiology*, 2018). As oridonin significantly inhibited both *cepl* and *rpfF_{BC}* gene expression (Fig.2), we then tested whether oridonin affects the expression of *rpfF_{BC}* and *cepl* through RqpR. These details have been described in line 136-140.

Fig. 3B. The EMSA results are not very clear. The free probe does not decrease with the addition of oridonin.

Response: Good suggestion, we have replaced the figure of Fig. 3B as suggested.

The literature reports several effects of Oridonin seems on many cellular process. It seems that many of these claims could be due to unspecific binding, protein aggregation, protein precipitation. Have the authors considered this possibility?

Response: Good suggestion. In this study, we have identified two direct targets of oridonin, RqpR and CepR (Fig.3 and Fig.4). However, it is possible that there are other unknown targets of oridonin, or other mechanisms employed by oridonin to

affect the functions of *B. cenocepacia*, which needs further investigation.

In order to show activity, most of the assays are performed at high concentrations(100uM). There is plenty of data on the toxic effects of oridonin. Have the authors look at the toxic effects at these concentrations?

Response: Thanks for your good suggestion. In this study, we found that oridonin could significantly reduce the production of quorum sensing signals, inhibit the biofilm formation, motility and protease activity of *B. cenocepacia* at a final concentration from 20-100 μ M (Fig.3, Fig.4 and Fig.5). In addition, we also found that oridonin remarkably attenuated *B. cenocepacia* virulence, while exerted nontoxic effect towards A549 cells at a final concentration from 12.5 to 100 μ M (Fig. 6).

Figure 6A and 6B. It is not clear how the authors calculated the percentages. The y-axis of figure 6B is labeled % WT. It is not clear what that means.

Response: Good suggestion, we have revised all the relevant y-axis of Figures in this manuscript as suggested.

The idea of targeting quorum sensing as an antivirulent target is not new. However, it did not so far render any promising results. The authors should exert caution on suggesting this approach as promising. Quorum sensing in Burkholderia is complex and not conserved in the different species. In addition, quorum sensing may not be relevant at chronic stages of infection.

Response: Good suggestion, we have revised the sentence as suggested (Page 1, Line 89). Our study just showed that oridonin inhibited QS of *B. cenocepacia* H111 and reduced virulence and inflammation caused by *B. cenocepacia* H111, but whether the quorum sensing systems play a role in the chronic infection of *B. cenocepacia* still needs further study.

Reviewer #2 (Comments for the Author):

This study showed the inhibitory efficacy of oridonin against two types of quorum sensing in the pathogenic *Burkholderia cenocepacia*. The various experimental data support the conclusions, and the manuscript is written in standard English and easy to comprehend. However,

1. Lack of description of the rationale for choosing oridonin as a QS inhibitor among all the other candidates.

Response: Good suggestion. Among these thirteen active candidate compounds, theaflavin-3,3'-digallate, thoningianin A, oridonin and acetyl-alpha-boswellic acid exhibited good inhibitory activity on both *rpfF_{BC}* and *cepl* gene expression. However, we have only identified the direct targets of oridonin. Isothermal Titration Calorimetry (ITC) analysis showed that only oridonin bound to RqpR with an estimated dissociation constant (K_D) of $8.28 \pm 0.895 \mu$ M (Fig.3, Fig. S2). So, we chose oridonin for further investigation in this study.

2. The concentration of 100 μ M, which showed effectiveness as a QS inhibitor, is pretty much high. Is it economical for practical use?

Response: Thanks for your good suggestion. In this study, we found that oridonin could significantly reduce the production of quorum sensing signals, inhibit the biofilm formation, motility and protease activity of *B. cenocepacia* at a final concentration from 20-100 μ M (Fig.3, Fig.4 and Fig.5). In addition, we also found that oridonin remarkably attenuated *B. cenocepacia* virulence, while exerted nontoxic effect towards A549 cells at a final concentration from 12.5 to 100 μ M (Fig. 6).

3. There is no discussion about the mode of action of oridonin as a QS inhibitor. Since the chemical structures of oridonin, BDSF and C8-HSL are different, it is less likely that the oridonin is a competitive inhibitor. It can easily get the evidence through a simple experiment to determine whether the oridonin is a non-competitive inhibitor. Please test whether increasing the concentration of C8-HSL does not affect biofilm, motility, or protease activity in the *cepl* mutant in the absence or presence of oridonin at the IC50 value.

Response: Good suggestion. We have added the biofilm formation experiments as suggested. Our results showed that BDSF and C8-HSL increased the biofilm formation of *rpfF_{BC}* and *cepl* mutants, respectively, in a dose-dependent manner in the absence of oridonin (Fig.S6, S7). Exogenous addition of 50 μ M BDSF and C8-HSL fully rescued the impaired biofilm formation of *rpfF_{BC}* and *cepl* mutants, respectively, in the absence of oridonin, while exhibited no any restored effects on the biofilm formation of the signal-minus mutants in the presence of 100 μ M oridonin (Fig.S6, S7). These results suggest the complicated action mechanisms and multiple targets of oridonin in *B. cenocepacia*, which needs further investigation. We have added these results in the revised version of our manuscript.

The following are minor points that are needed to improve the manuscript.

1) Lines 97, 215: "Burkholderia" should be italic.

Response: We have revised it as suggested.

2) Lines 111-113: data not shown?

Response: We have revised it as suggested.

3) Lines 118-121: The levels of cytotoxicity of bilirubin look similar to its theaflavin 3'3-digallate. Please clarify the percentage of cytotoxicity as the baseline for choosing the compounds.

Response: Thanks for your good suggestion. We have modified this sentence.

4) Lines 128-133: Based on your data (Fig. 2A), β -hydroxyisovalerylshikonic acid is not an effective inhibitor against *rpfF* gene expression. Same as the theaflavin 3'3-digallate, protopseudohypericin, ginsenoside Rk1, theaflavin-3-digallate, and α -boswellic acid. Please consider re-select the compounds which show the significant differences.

Response: Thanks for your good suggestion, we have revised this sentence as

suggested.

5) Lines 200-201: Please explain the reason in the Discussion.

Response: Thanks for your good suggestion, we have revised it as suggested.

6) Methods: If you used a commercial product of oridonin, please mention the company of the product.

Response: Thanks for your good suggestion. We have added more information of oridonin in the Methods section as suggested.

7) Figure 4B and lines 173-175: The intensities of the band of the bound probe are pretty much the same between 20 μ M and 10 μ M of oridonin. It seems that the amount of the bclACB promoter probe bound to the CepR did not decrease dependent on increasing oridonin concentration.

Response: Good suggestion, we have repeated the EMSA experiments and replaced the picture.

8) Method: In general, QS signal receptor proteins are insoluble when overexpressed without the cognate signals. Please describe the details of the purification process of signal receptor proteins, and indicate whether the QS signals were added to the EMSA reaction.

Response: Good suggestion, we have added more information in the Methods section as suggested.

July 4, 2022

Prof. Yinyue Deng
Sun Yat-sen University
Guangzhou 510642
China

Re: Spectrum01787-22R1 (Oridonin attenuates *Burkholderia cenocepacia* virulence by suppressing quorum sensing signaling)

Dear Prof. Yinyue Deng:

Your manuscript has been accepted, and I am forwarding it to the ASM Journals Department for publication. You will be notified when your proofs are ready to be viewed.

Sincerely,

Giordano Rampioni
Editor, Microbiology Spectrum
